# Dynamics of valence-shell electrons and nuclei probed by strong-field holography and rescattering

Samuel G. Walt[1], Niraghatam Bhargava Ram[1], Marcos Atala[1,†], Nikolay I. Shvetsov-Shilovski[2], Aaron von Conta[1], Denitsa Baykusheva[1], Manfred Lein[2] & Hans Jakob Wörner[1]

Strong-field photoelectron holography and laser-induced electron diffraction (LIED) are two powerful emerging methods for probing the ultrafast dynamics of molecules. However, both of them have remained restricted to static systems and to nuclear dynamics induced by strong-field ionization. Here we extend these promising methods to image purely electronic valence-shell dynamics in molecules using photoelectron holography. In the same experiment, we use LIED and photoelectron holography simultaneously, to observe coupled electronic-rotational dynamics taking place on similar timescales. These results offer perspectives for imaging ultrafast dynamics of molecules on femtosecond to attosecond timescales.

[1] Laboratorium für Physikalische Chemie, ETH Zürich, Vladimir-Prelog-Weg 2, HCI E 237, 8093 Zürich, Switzerland. [2] Institut für Theoretische Physik, Leibniz Universität Hannover, 30167 Hannover, Germany. † Present address: Max Planck Institute for the Structure and Dynamics of Matter, Center for Free Electron Laser Science, Luruper Chaussee 149, 22761 Hamburg, Germany. Correspondence and requests for materials should be addressed to H.J.W. (email: hwoerner@ethz.ch).

Time-resolved measurements of molecular dynamics have made substantial progress over the past years. In particular, electronic dynamics in atoms and molecules have become accessible through recent developments in attosecond and strong-field science[1–10]. This fundamental progress has so far remained restricted to methods relying on spectroscopy, rather than diffraction. In other words, electronic dynamics have been recorded by measuring either the time-dependent populations of electronic eigenstates or their accumulated phase differences, which does not provide any structural information about electronic wave functions. However, the interaction of a molecule with a strong laser field naturally provides access to structure by inducing rescattering between the laser-driven photoelectron wave packet and the parent ion. This so-called laser-induced electron diffraction (LIED)[11,12] has been used to probe the structure of static atoms and molecules[13–20], and to obtain evidence of nuclear dynamics following strong-field ionization[21]. Photoelectron holography, which results from interference between electrons that have scattered from the parent ion and electrons that have not, has also been used to interrogate the structure of static atoms and molecules[22–24], and to reveal nuclear dynamics following ionization[25]. Although the imprint of electronic dynamics on photoelectron distributions has been reported for neutral atoms[26], it is not obvious whether the distributions can be interpreted in terms of LIED or holography. As far as molecules are concerned, electronic dynamics have so far escaped observation by any of these powerful methods. As LIED and holography rely both on the wave nature of the electron and on rescattering, they are expected to be sensitive to small variations of the molecular structure. Earlier work on electron interference without rescattering[27–30] has already revealed the sensitivity to the symmetry of the bound state.

In this study, we transpose photoelectron holography and LIED from static systems to probing coupled electronic and nuclear dynamics in molecules. We first concentrate on purely electronic dynamics and show that a valence-shell electron wave packet leads to a very strong contrast modulation of the holographic fringes. Our calculations trace the origin of this effect to the time dependence of the momentum-space electron wavefunction, which modulates the amplitude ratio and the relative phases of scattering and non-scattering trajectories. We then investigate the manifestation of coupled electronic and nuclear dynamics taking place on similar time scales. We find signatures of both types of dynamics in photoelectron holography, but LIED is found to be almost exclusively sensitive to the nuclear dynamics. These results suggest avenues for disentangling electronic and nuclear dynamics in molecules, which is particularly interesting in the case of non-adiabatic dynamics, such as those occurring at conical intersections[31].

## Results

**Experiment**. We use impulsive stimulated Raman scattering to prepare a coupled electronic and rotational wave packet in the neutral NO molecule. A supersonic molecular beam was formed by expanding a 1% mixture of NO with helium through a pulsed Even–Lavie valve. Several rotational states within the $^2\Pi_{1/2}$ ground electronic state and the $^2\Pi_{3/2}$ first excited electronic state (separated by 15 meV) are impulsively prepared by a non-resonant laser pulse (800 nm, 145 fs, $4 \times 10^{13}$ W cm$^{-2}$ peak intensity). The long duration of the laser pulse is chosen to avoid the preparation of vibrational wave packets in NO and NO$^+$ (periods of 17 and 14 fs, respectively). The wave packet is subsequently probed by strong-field ionization (800 nm, 35 fs, $2.3 \times 10^{14}$ W cm$^{-2}$) and the photoions and photoelectrons are detected by a velocity-map-imaging spectrometer[32,33]. We also

recorded ion time-of-flight measurements (Fig. 1a) by adding <1% Xe to the NO/He mixture and the NO$^+$ signal was normalized to the Xe$^+$ signal, to eliminate fluctuations of laser parameters and gas-jet density.

Figure 1a shows the normalized NO$^+$ yield as a function of the pump–probe delay. The rapid regular oscillation originates from the electronic dynamics, illustrated in the inset, whereas the two features around 5 and 10 ps correspond to rotational revivals. The amplitude modulations observed around 8–9.5 ps are a characteristic signature of multi-level quantum beats[34,35]. Figure 1b shows a cut through the three-dimensional photoelectron momentum distribution recorded at a delay of 1.56 ps. Details about the processing of the photoelectron momentum images are given in the Methods section. Figures 1c,d show normalized differences, defined as $S(t_1, t_2) = 2(D(t_2) - D(t_1))/(D(t_2) + D(t_1))$ of photoelectron momentum distributions $D$ recorded at the local minimum (delay $t_1$) and maximum (delay $t_2$) of the photoelectron (or NO$^+$) yield that are dominated by either purely electronic or electronic and rotational dynamics, respectively. The corresponding values of $t_1$ and $t_2$ are given in the caption of Fig. 1. The purely electronic valence-shell dynamics (Fig. 1c) appears as a pronounced modulation of a holographic pattern, whereas the rotational dynamics manifests itself by additionally modifying the angular distribution of rescattered photoelectrons, clearly visible at the highest momenta of Fig. 1d.

**Electronic dynamics**. We first concentrate on the purely electronic dynamics. Figure 2a (reproduced from Fig. 1c) shows stripes of alternating orange and yellow colours extending from the centre towards the border of the momentum distribution. We attribute these structures to photoelectron holography, previously observed for stationary states of atoms[22,23,36]. This assignment is substantiated by comparing the observed fringe pattern with the quantitative prediction of holography. Using the vertical ionization potential of NO and the conditions of our experiments leads to the prediction of regions of constructive (black) and destructive (white) interference calculated following the simple model outlined in ref. 36. These fringes arise from the interference between quantum trajectories of (signal) electrons that have scattered forward from the parent ion and (reference) electrons that have passed by the parent ion without scattering. The lack of quantitative agreement between the experiment and the simple model mainly originates from the absence of the Coulomb potential in the latter, as we further discuss below.

The sudden loss of fringe visibility with increasing momentum constitutes further evidence of the holographic origin of the fringes. The green circle in Fig. 2a marks the maximal momentum that can be acquired by electrons that do not scatter. The blue line represents the $1/e^2$ full width of the tunnelling filter, defined by

$$G\left(p_x^2 + p_y^2\right) = e^{-\frac{\left(p_x^2 + p_y^2\right)\tau}{2}},\tag{1}$$

where $p_x$ and $p_y$ are momentum components perpendicular to the laser-field direction and $\tau = \sqrt{2I_p}/F$, where $I_p$ represents the ionization potential and $F$ is the instantaneous field strength at the moment of ionization. The fringes are only observed within the area defined by the green circle, showing that they involve non-scattered electrons. More precisely, we find that the fringes are even limited to the area within the blue curve, showing that the maximal lateral momentum of the electron wave packet is indeed limited by the tunneling filter, lending support to the concept proposed in refs 15,37. We emphasize that the

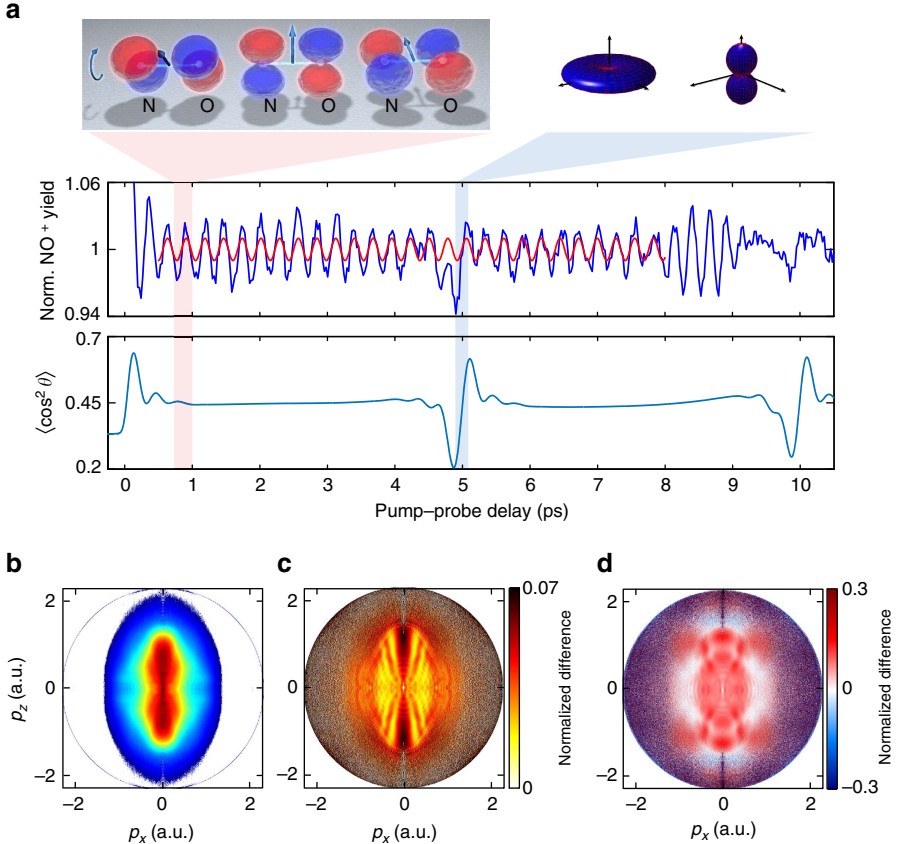

**Figure 1 | Discerning electronic and nuclear dynamics.** (**a**) The top panel illustrates the prepared electronic and rotational dynamics with the left part showing the time-evolution of the one-electron wave function in NO and the right part illustrating the molecular-axis distribution. The wave function can be chosen real-valued with opposite signs (red and blue) by multiplication with a global phase factor that corresponds to a physically irrelevant shift of the absolute energy scale. The middle panel shows the time-dependent normalized $NO^+$ yield (blue line) and a sine function with a period of 277 fs (red line). The lowest panel shows the calculated alignment dynamics expressed by $\langle\cos^2\theta\rangle$ (including volume averaging). (**b**) Photoelectron momentum distribution of excited NO molecules recorded at a delay of 1.56 ps. Normalized difference $S(t_1, t_2)$ (defined in the text) of momentum distributions measured at the minimum and maximum of the $NO^+$ signal dominated by electronic dynamics (**c**, $t_1 = 1.56$ ps and $t_2 = 1.72$ ps, respectively) or around the rotational revival (**d**, at $t_1 = 4.91$ ps and $t_2 = 5.10$ ps, respectively). a.u., atomic units.

holographic fringes observed in Fig. 2a are barely visible in the inverted photoelectron momentum distributions themselves (Fig. 1b). They only become visible in the normalized differences. Figure 2b shows how the contrast of the holographic fringes evolves in time over one period of the electronic dynamics. The holographic fringes in the normalized-difference images modulate with a contrast close to 100%.

A weak signature of the electronic dynamics can also be observed outside the green circle of Fig. 2a, that is, in the region dominated by the electrons that have scattered backwards from the parent ion. We note that the noise along the central vertical line is an artifact of the inversion procedure. The nearly constant value of the difference signal observed in this region shows that the electronic dynamics does not change the angular distribution function of the rescattered electrons. However, the total signal of the rescattered electrons modulates in time with approximately the same contrast as the total signal. We thus conclude that the electronic dynamics in NO causes a variation of the rescattering probability, but it hardly affects the shape of the photoelectron distribution in the rescattering region. In contrast, the shape of the holographic pattern shows a strong signature of the electronic dynamics.

**Model for time-dependent photoelectron holography.** We now introduce a model for time-dependent photoelectron holography and use it to explain our observations. Figure 3a shows the temporal

evolution of the time-dependent orbital describing the unpaired electron of NO. The displayed orbital wavefunction corresponds to one of two possible values of the spin projection quantum number ($\Sigma = +1/2$). The orbital corresponding to the opposite value of $\Sigma$ rotates in the opposite direction[8] and is omitted here for clarity. As holography results from the interference of scattered and non-scattered trajectories, we focus on the momentum distribution of electrons perpendicular to the direction of tunnelling, also called lateral momentum distribution. We determine this quantity for the time-dependent electronic wave packet by building on the concept of partial-Fourier transformation for strong-field ionization[38]. The laser field is linearly polarized along the $z$ axis. A cut through the one-electron density in the $xy$ plane at the distance $z_0 = -1.83$ Å from the origin is shown on the front face of each cube in Fig. 3a. $z_0$ corresponds to the position of the local maximum of the combined Coulomb and laser-field potentials using the peak electric field strength. The two-dimensional Fourier transform of the wave function at $z = z_0$, multiplied by the tunnelling filter (equation 1) leads to the lateral-momentum distribution of the continuum photoelectron wave packet after tunnelling, shown in Fig. 3b:

$$\Phi(p_x, p_y, z_0, t) \approx \left| \frac{1}{2\pi} \iint \Psi(x, y, z_0, t) e^{ixp_x + iyp_y} \mathrm{d}x\mathrm{d}y \right. $$
$$\left. \times G\left(p_x^2 + p_y^2\right) \right|^2. \tag{2}$$

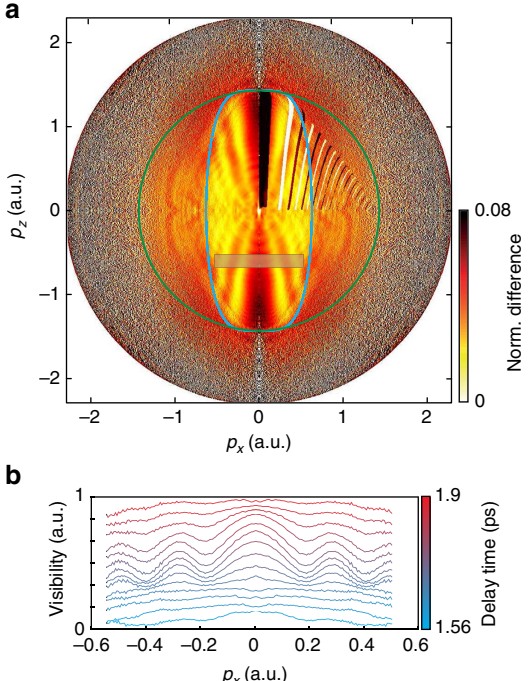

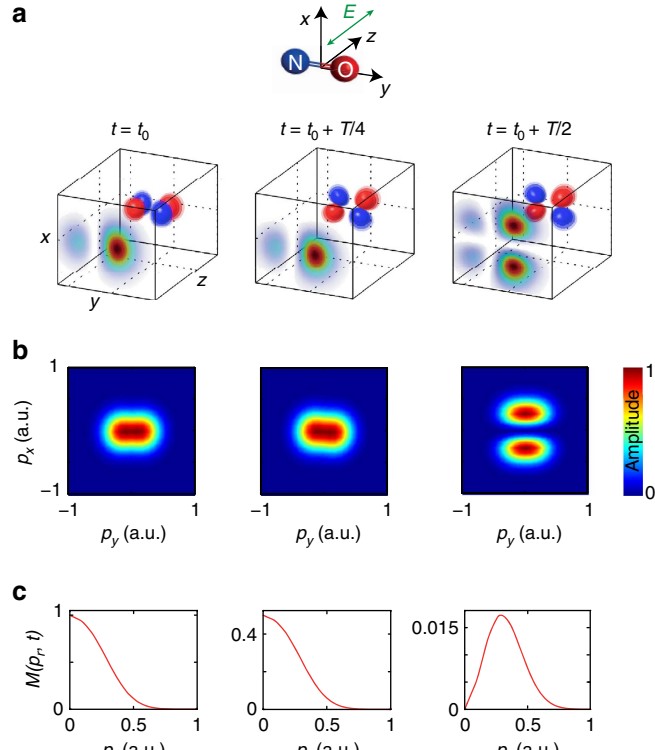

**Figure 2 | Time-dependent photoelectron holography.** (**a**) Normalized difference $S(t_1, t_2)$ between delays $t_1 = 1.56$ ps and $t_2 = 1.72$ ps, dominated by electronic dynamics. The added curves are discussed in the text. The polarization of both laser pulses was parallel to the $p_z$ axis. (**b**) Lineouts of the holographic pattern obtained by integrating the signal inside the grey box in **a** over $p_z$, but recorded at the indicated delays that sample one period of the electronic dynamics and are referenced to a delay of 1.05 ps.

The corresponding angle-averaged distribution function $M(p_r, t)$, shown in Fig. 3c, is defined by

$$M(p_r, t) = \frac{\Phi'(p_r, z_0, t)}{\Phi'(p_r = 0, z_0, t = t_0)}, \tag{3}$$

where

$$\Phi'(p_r, z_0, t) = \frac{\oint_{p_r} \Phi(p_x, p_y, z_0, t) \, ds}{\oint_{p_r} ds} \tag{4}$$

and $p_r^2 = p_x^2 + p_y^2$.

The time-evolving lateral-momentum distribution shown in Fig. 3b has a transparent physical interpretation. As time evolves, the orbital wave function of the unpaired electron shown in Fig. 3a rotates around the molecular axis, leading to the periodic appearance and disappearance of a nodal plane containing the direction of tunnelling ($z$).

We further support the interpretation of the observed holographic fringes by turning to classical-trajectory calculations[39] that are described in more detail in the Methods section. We consider the two cases where the one-electron density is aligned either parallel to the polarization direction of the ionizing laser pulse ($t = t_0$) or perpendicular to it ($t = t_0 + T/2$). We propagate classical trajectories in the combined Coulomb and laser field, using the experimental parameters for the latter. The initial momenta sample the distribution function derived above for the highest-occupied molecular orbital of NO obtained from quantum-chemical calculations using the GAMESS package. Figure 4 shows the calculated final momentum distributions at the detector for (a) the electron density being aligned along the direction of the ionizing laser polarization, (b) the electron density being aligned perpendicular to this direction and (c) the

**Figure 3 | Model for time-dependent photoelectron holography.** (**a**) Temporal evolution of the one-electron wave function of NO and cut through the density along the vertical plane located at the maximum of the laser + Coulomb potential. (**b**) Lateral-momentum distributions obtained by 2D-Fourier transformation of the wave-function cuts shown in **a** and multiplication with the tunnelling filter. (**c**) Distribution function of lateral momentum norms $M(p_r, t)$ of **b**.

normalized difference of (a) and (b). The momentum distribution (a) shows the characteristic features of strong-field ionization by a near-infrared laser pulse, that is, the above-threshold ionization peaks, which extend up to $2U_p$ and the holographic fringes. The momentum distribution (b) is significantly weaker in amplitude (note the different colour scales), broader in the lateral-momentum direction and additionally displays a node at $p_x = 0$. This feature reflects the nodal plane of the molecular orbital, which contains the laser polarization in this specific configuration. This feature is not visible in the experimental data, because the fraction of excited molecules is on the order of 1% (refs 34,35) and the molecules are not aligned at the considered delays, whereas 100% excitation and perfect alignment was assumed in the calculations.

Figure 4c shows the normalized difference of Fig. 4a,b and is in excellent agreement with the experimental result (Fig. 2a). In particular, the calculation nicely displays all characteristic properties of the experimentally observed holographic structures. Interestingly, the comparison of these calculations with the simple model from ref. 36 shows the same discrepancies as Fig. 2a, that is, the model overestimates the spacing of the holographic fringes along the lateral-momentum direction. The improved agreement between the experiment and our trajectory calculations must therefore originate from the inclusion of the Coulomb potential in the latter.

These complete three-dimensional trajectory calculations further show that the electronic dynamics indeed modulate the contrast of the holographic fringes. As the calculations were done using a purely Coulombic potential, the observed time-dependent

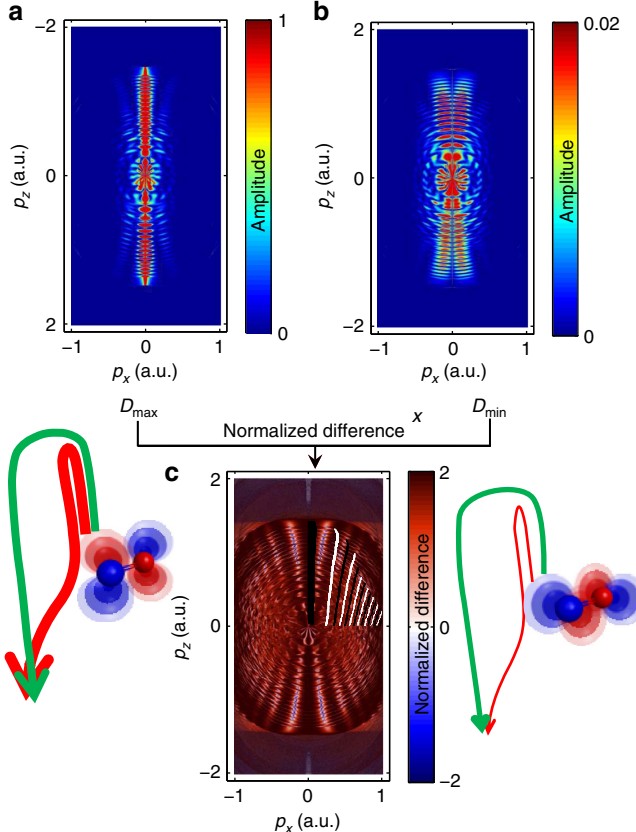

**Figure 4 | Classical trajectory calculations.** Calculated momentum distribution for the cases where the electron density of the highest-occupied molecular orbital (HOMO) is aligned (**a**) along the polarization direction of the ionizing laser field or (**b**) perpendicular to it. (**c**) The normalized difference of the momentum distributions shown in **a,b**.

holography can be uniquely attributed to the time-dependent momentum-space structure of the outermost orbital of NO as illustrated in Fig. 3. The time dependence of the orbital is translated into a time-dependent lateral-momentum distribution of the photoelectron wave packet after tunneling, which results in a high contrast modulation of the holographic fringes. The good agreement between theory (Fig. 4c) and experiment (Fig. 2a) also shows that the molecular properties of the potential seen by the continuum electron are not crucial in defining the observed holographic pattern. This insensitivity may turn out to be an advantageous filter that makes time-dependent photoelectron holography specifically sensitive to a particular aspect of the electronic dynamics that it records, that is, the lateral momentum distribution.

**Coupled electronic and nuclear dynamics.** We now turn to the observation of coupled nuclear and electronic dynamics occurring around a rotational revival. Figure 1a compares the measured $NO^+$ signal with calculations of the rotational dynamics of NO. At the early delays discussed so far (1–4 ps), rotational dynamics can be neglected, because the axis distribution does not change over the period of the electronic dynamics. In contrast, electronic and nuclear dynamics occur on similar time scales around a rotational revival, which offers the opportunity to study the manifestation of coupled electronic and nuclear dynamics. Figure 5a compares the photoelectron yield integrated over all momenta around the revival of the rotational dynamics (top) with the extrapolated electronic quantum beat (centre) and the

calculated rotational dynamics (bottom) taken from Fig. 1a. These calculations have been done with the methods described in refs 34,35. They include the rotational levels of both populated electronic states of NO and fully account for the coupled electronic and rotational dynamics on the relevant time scales.

Figures 5b–d show normalized differences of photoelectron momentum distributions $S(t_2, t_1)$, $S(t_2, t_3)$ and $S(t_2, t_4)$, respectively. These delays were chosen to highlight the different manifestations of coupled electronic and nuclear dynamics. The delays $t_1$ and $t_2$ both lie close to the minimum of $\langle \cos^2\theta \rangle$, which means that the axis distribution undergoes little change within this time interval. However, the interval $[t_1, t_2]$ samples a significant fraction of the electronic quantum beat. Between $t_2$ and $t_3$, both electronic and nuclear dynamics evolve. Between $t_2$ and $t_4$, the electronic dynamics have evolved for half of a period and the rotational dynamics have undergone their maximal evolution, from anti-alignment to alignment.

We first note that the minimum of the photoelectron and $NO^+$ yield ($t_2$) lies between the minima of $\langle \cos^2\theta \rangle$ (close to $t_1$) and that of the extrapolated purely electronic quantum beat (close to $t_3$). This is a first indication that both electronic and nuclear dynamics influence the ionization probability. Much more details are visible in the photoelectron images. We focus our discussion on three aspects of these images. First, Fig. 5b–d all display holographic signatures similar to those of Fig. 2a. These features coincide with the location of the measured main holographic maxima in Fig. 2a, which are reproduced here as black dashed lines. The excellent agreement shows that the structures observed in Fig. 5b–d also correspond to photoelectron holography. These holographic fringes are not unexpected because all three intervals sample substantial fractions of an electronic quantum beat.

Second, each of these six main holographic fringes displays an additional extremum in the radial dimension, which was not observed in Fig. 2a. Comparing Fig. 5b with Fig. 5c,d, we observe a maximum along the radial dimension in the case of Fig. 5b and a minimum in Fig. 5c,d. These radial extrema cannot be explained by purely electronic dynamics because of the differences observed in comparison with Fig. 2a. We note that normalized-difference images evaluated between delays sampling fractions of the purely electronic quantum beat in the range of 1–2 ps all look qualitatively identical to Fig. 2a (not shown). The additional structures on top of the holographic fringes can therefore only originate from the simultaneous effect of electronic and nuclear dynamics, which is thus found to leave a characteristic imprint on time-dependent photoelectron holography.

Third, Fig. 5d additionally shows pronounced structures in the region of the rescattered photoelectrons (outside the green circle). This shows that the distribution shape of the rescattered electrons with energies above $2U_p$ is also modulated, which was clearly not observed during the purely electronic dynamics (compare Figs 2a and 5d). We further investigate this aspect by analysing the rescattered electrons. Outside the green circle in Fig. 5d, alternating regions of negative (blue) and positive (red) colour are observed. We analyse this region using the LIED approach illustrated with a black circle and arrow[21,40,41]. Electrons that have elastically scattered with the momentum $|\mathbf{p}_s|$ in the direction $\alpha$ acquire the additional momentum $\mathbf{p}_L = -\mathbf{A}(t_r)$ from the laser field, equal to the opposite value of the vector potential at the moment of rescattering ($t_r$). The value of the normalized difference along such circles is displayed in Fig. 5f and compared with calculations (Fig. 5e). We use the variational Schwinger method implemented in ePolyScat[42,43] to accurately solve the scattering problem of electrons from aligned $NO^+$ molecules. Our calculations also take into account the angle-dependent strong-field ionization rate of NO[44] and the

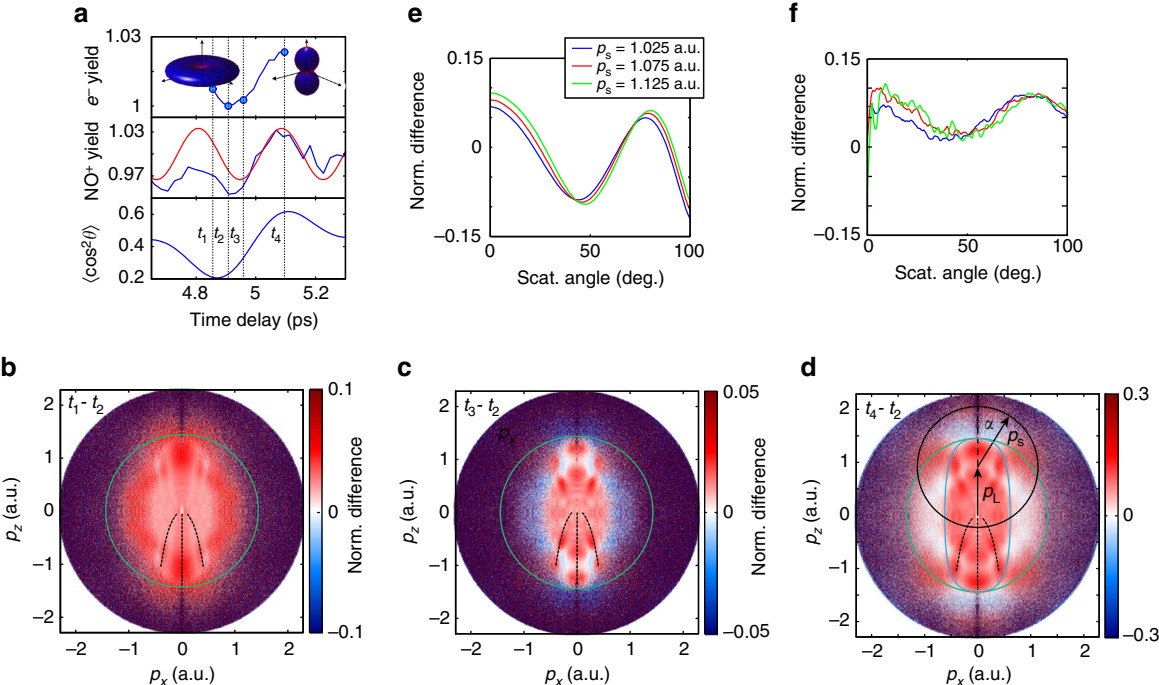

**Figure 5 | Manifestation of coupled electronic-nuclear dynamics in holography and LIED.** (**a**) Normalized photoelectron yield integrated over all momenta around the first revival (top), $NO^+$ yield in blue and extrapolated electronic quantum beat in red (middle) and calculated alignment dynamics (bottom). (**b**) Normalized difference $S(t_2, t_1)$ for $t_1 = 4.86$ ps and $t_2 = 4.91$ ps. The green circle indicates the $2U_p$ limit. The black dashed lines indicate the position of the maximum of the holographic pattern of Fig. 2a. (**c**) $S(t_2, t_3)$ for $t_3 = 4.96$ ps. (**d**) $S(t_2, t_4)$ for $t_4 = 5.10$ ps, the blue line is the same as in Fig. 2a. The black circle indicates the final momentum of photoelectrons with a scattering momentum $|p_s|$ and scattering angle $\alpha$ ($\alpha = 0$ corresponds to back-scattering). (**e**) Calculated normalized difference of rescattered electron distributions for different scattering momenta. (**f**) Experimental normalized differences as a function of the scattering angle for the same scattering momenta as in **e**, smoothed over a 10° range, using the same colour coding as in **e**.

molecular-axis distribution corresponding to our experimental conditions. The prediction shown in Fig. 5e qualitatively agrees with the experimental result in Fig. 5f. Both the general shape of the curves and the shift of the minimum to larger angles for increasing scattering momenta are well reproduced. The calculation however overestimates the contrast of the normalized difference, which can originate from the limited accuracy of the electron-molecule scattering calculations and/or the strong-field ionization rates. We have verified that uncertainties in the molecular axis distributions cannot explain the observed discrepancy.

## Discussion

We have reported on the observation of valence-shell electron and coupled electronic-nuclear dynamics using strong-field photoelectron holography and rescattering. We will separate our conclusions into two parts, that is, conclusions that we expect to be general and conclusions that we view as specific to the broad class of radicals (such as NO, $NO_2$, OH and so on). We have shown that photoelectron holography can be used as a probe of valence-shell dynamics in molecules, and that it is particularly sensitive to the components of the momentum wave function perpendicular to the direction of tunnelling. These components indeed control the relative probability and phase of rescattering relative to non-rescattering trajectories, which directly control the contrast and structure of the holographic pattern, respectively. We have further shown that coupled electronic and nuclear dynamics leave a characteristic imprint on the holographic pattern. This suggests the application of photoelectron holography to probe coupled electronic and nuclear dynamics in polyatomic molecules. We expect these two conclusions to be general. As NO has a single unpaired electron, we expect some of

our observations to apply to radicals. Specifically, we have observed that the electronic dynamics leave no signature on the angular-distribution shape of the rescattered electrons (see Fig. 2a), such that LIED provides a virtually background-free measurement of nuclear dynamics. This observation constitutes an experimental demonstration of the fundamental quantum-mechanical principle that the laser-driven electron wave packet must be described as scattering from its parent ion, rather than from the neutral molecule. In systems featuring a single unpaired electron, such as NO, the parent ion does not display electronic dynamics, resulting in a delay-independent LIED. In the more frequent case of closed-shell molecules, the returning electron will scatter off an ion supporting electronic dynamics, such as charge migration[10]. Our results suggest that in this case, the electronic dynamics of the neutral molecule will be observable in holography, which will however additionally contain signatures of the ionic dynamics (both electronic and nuclear), contributed by the rescattering quantum trajectories. In contrast to holography, LIED will mainly reflect the dynamics of the ion, because the angle dependence of the rescattered electrons is dominated by the elastic-scattering cross-section[41]. More broadly, these considerations show how the fundamental principles uncovered in our pump–probe experiment can be transferred to the regime of sub-cycle temporal resolution to observe, for example, coupled structural and electronic rearrangements on femtosecond to attosecond timescales.

## Methods

**Data processing.** The two-dimensional photoelectron momentum images were processed as follows. A constant background was subtracted from the entire image to account for dark counts and residual stray light. Then, the images were symmetrized. Finally, two-dimensional momentum distributions were obtained by iterative Abel inversion using a computer program developed in our group[33].

A typical photoelectron momentum distribution is shown in Fig. 1b. The momentum distributions shown in Figs 1c,d,2a and 5b–d are normalized differences between distributions measured at two different pump–probe delays, as defined in the text. The analysis of the LIED signals proceeded as follows. For a given photoelectron momentum at rescattering $\mathbf{p}_s$, the corresponding time of rescattering $t_s$ was determined using the long trajectories in the classical recollision model (see, for example, ref. 41). The short trajectories were discarded, because they are suppressed by the lower strong-field ionization rate at the corresponding times of ionization. The additional momentum shift $\mathbf{p}_L$ that the electrons acquire from the laser field after rescattering was calculated according to $\mathbf{p}_L = -\mathbf{A}(t_r)$, using the vector potential corresponding to the laser field used in our experiment.

**Trajectory calculations.** We calculate photoelectron momentum distributions for different times during the electronic wave-packet dynamics using the semiclassical two-step model for strong-field ionization[39], which is a Monte-Carlo method involving interfering trajectories. Electron trajectories are launched throughout the laser pulse with probabilities corresponding to the instantaneous tunnel ionization rate. As initial conditions we use the lateral momentum distributions derived above (see equation 3) in combination with zero initial momentum along the field and initial electron position at the tunnel exit of a triangular barrier determined by the instantaneous electric field. As the method[39] is based on Newtonian trajectories moving in a cylindrically symmetric potential formed by the laser field and a Coulomb potential, the simulation can be restricted to trajectories in a plane containing the field direction. However, electrons departing towards opposite sides at the time of ionization may end up with the same final momentum and interfere. Thus, initial conditions for both negative and positive lateral momenta must be defined. The lateral distribution is chosen to be symmetric in both cases that we consider, $t = t_0$ and $t = t_0 + T/2$. In the former case, the initial phases are taken equal on both sides, but in the latter case a $\pi$ phase difference is used, in view of the orbital symmetry (see Fig. 3a,b). Similarly, in the former case, the initial phases have a $\pi$ difference for opposite directions of the ionizing field and they are taken equal for both field directions in the latter case. The laser parameters for the simulation are the experimental ones except that a short $\sin^2$-shaped envelope with total length of eight optical cycles is used instead of the longer experimental pulses.

**Data availabilty.** The data that support the findings of this study are available from the corresponding author upon reasonable request.

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

## Acknowledgements

We gratefully acknowledge funding from the Swiss National Science Foundation (Project Number 200021_138158 and the NCCR-MUST), ETH Zürich and an ERC Starting Grant (Project Number 307270-ATTOSCOPE). N.B.R. acknowledges support from a

Swiss Government Excellence Scholarship. N.S.-S. and M.L. acknowledge the support within the QUEST-Leibniz Research School Hannover.

## Author contributions

S.G.W. and H.J.W. conceived and designed the experiments. S.G.W., N.B.R. and A.v.C. performed the experiments and analysed the data. S.G.W., D.B. and M.A. developed the model calculations and the rescattering calculations. N.I.S.-S. and M.L. implemented the semiclassical-trajectory calculations. All authors contributed to writing the manuscript.

## Additional information

**Competing interests:** The authors declare no competing financial interests.

**Publisher's note**: 

