## [Peer Review File · Nature Communications]

Reviewers' comments:

Reviewer #1 (Remarks to the Author):

In this manuscript, the authors study electronic and nuclear dynamics in NO molecule, by means of laser-induced electron diffraction (LIED). This powerful technique was previously used to study either static atomic/molecular systems or nuclear dynamics in molecules. Here the authors are interested in observing the coupling between both "motions" in the region of alignment revival.

The manuscript comprises two parts. They first look at the molecule when there is no "fast" nuclear motion and could determine, with the LIED, the signature of the purely electronic motion. A comparison with theory is in rather good agreement with the experimental results. In the second part, they concentrate on the "coupled region" and find strong modifications of the measured signals due to the interplay between the dynamics. Once again theoretical results support the experimental observations.

While the sum of the results presented here are very interesting for the community, I cannot recommend this manuscript for publication in Nature Communications in its present form for the following reasons.

1/ I think that the techniques presented here lack the novelty character. Both on the side of theory (the use of classical trajectories to simulate the diffraction signal) or experiment (impulsive alignment of molecules and holography), they were developed and presented elsewhere by other teams. The originality of this work is mainly contained in its subject and the results are convincing.

2/ I believe that this manuscript is not really well written. It is sometimes hard to follow the flow of thinking and some informations are scattered in different parts. For example it is difficult to understand how the results of Fig. 2b are obtained. I suggest that the authors define a quantity $S(t_1, t_2) = [2(D(t_1) - D(t_2))] / (D(t_1) + D(t_2))$, and use it throughout the paper to characterize the quantities plotted.

3/ I have few questions concerning some results presented here.

a/ As the electronic oscillation period is 270fs, I do not understand why the difference in time between the max (1.72ps) and the min (1.56ps) is 160fs corresponding to a period of 320fs? The same applies to the other timings with a difference of 190fs.

b/ In Fig. 1a, The authors schematically represent the oscillation of the wavefunction and I suppose that the color coding would correspond to its sign. I think it is misleading as it is the density which is oscillating (rotating) and not the wavefunction or the latter acquires a phase and becomes complex. I think the authors should be more precise on that point. The same remark applies to Fig. 3a.

c/ In Fig. 2a, the authors added the white and black lines to indicate the max and the min of the interferences. While the agreement is good for the two first lines, there is a shift for the others. Is there a simple explanation? I believe the authors should comment on that point. Furthermore to increase the visibility of these fringes for the reader, I would suggest to the authors to add these black/white lines, only on the left part of the figure.

d/ I am troubled by the 2nd paragraph of p.3. The authors claim that there is almost no angular variation of the signal in Fig2.a in this region, where I can clearly see stripes with changes of around 50% (red and black). Is this due to the detection process? Furthermore, I do not understand the conclusion of this paragraph, and think that the authors should develop a little their point.

e/ In Fig. 4, the authors present holography signal from classical trajectory calculations. It is very hard to quantify the agreement with the experiment, in particular the periodicity of the fringes, as they are not shown in the same frame.

f/ Commenting Figs. 5, the authors state that the agreement between theory (e) and experiment (f) is very good. While I agree on the shape of the curves, I seriously question the way they present their results. In particular the abscissa range are not the same!! It is from -0.2 to 0.2 for the simulations and -0.05 to 0.15 for the experiments. The problem is that by making this choice, the reader miss the fact that the curves never cross zero in the experiments which they do in the simulations. I would

prefer that the authors use the same frame and comment on the discrepancy eventhough there would be a qualitative agreement.

g/ I could not find the values of t1,t2,t3 and t4 in the manuscript.

Reviewer #2 (Remarks to the Author):

Referee report on the manuscript NCOMMS-16-29217-T: 'Dynamics of valence-shell electrons and nuclei probed by strong-field holography and rescattering' by Samuel G. Walt et al.

The authors report on an experimental study where they investigate the electronic and nuclear (rotational) dynamics that are taking place in NO molecules subjected to a strong femtosecond laser pulse. They use an electron imaging spectrometer operated in the velocity mapping mode to measure the final momentum distribution of the photo-electrons as a function of the delay between a pump pulse that initiates the dynamics and a probe pulse that photo-ionizes the NO molecule.

By investigating the normalized difference of momentum distributions recorded at different pump probe delays, the authors demonstrate that, depending on the delay, the observed (holographic) interference pattern is dominated either by the electronic dynamics alone, or by the coupled electronic and rotational dynamics.

The manuscript is well written, and the results include some novel and interesting aspects. My main criticism is that earlier original work on the subject is not properly cited:

A. In their work, the authors use essentially the same experimental technique and methods as in Refs [PHYSICAL REVIEW A 87, 031404(R) (2013)] and [PHYSICAL REVIEW A 88, 023423 (2013)], where the electron motion in the ground state of the carbon atom, and the dynamics of orbital alignment in the ground state of several atoms were time resolved. In view of these earlier studies (that have to be cited here), the authors need to revise one of their main claims, that is that holographic imaging has remained restricted to static systems and has never been applied before to the study of (bound) electronic dynamics. The new aspect of the present study thus only resides in the identification of the *coupled* electronic and nuclear motion.

B. I also would like to bring to the authors attention that (holographic) interference of different electron trajectories emerging on either side of an atom or ion during strong field photo-ionization was predicted and discussed theoretically long time ago (see for instance Ref. [Phys. Rev. A 55, 3760 (1997)]). It was shown there, how the symmetry of the electron wave function determines the relative phases of the interfering trajectories. Experimentally, the first direct experimental observation of the interference effect (i.e. the holographic interference pattern), succeeded in experiments on strong field photo-detachment of negative ions in Ref. [Phys. Rev. Lett. 87, 243001 (2001)] (i.e. 10 years before the publication of Ref. [22]). The interference effect and its dependence on the symmetry of the initial state have been investigated in detail in a series of studies (see for instance Refs [J. Mod. Opt. 50, 461, (2003)] and [Phys. Rev. A 75, 063415 (2007)]). Even though a different nomenclature was used, (i.e. the term 'interference effect' rather than the term 'holography'), the effect is the same. These original studies have to be properly cited in the context of strong field electron holography.

Having pointed this out, I find the results of the present study particularly interesting. By extending the methodology of Ref. [PHYSICAL REVIEW A 87, 031404(R) (2013)] to a molecular target, the authors could investigate the effect of *coupled* electronic and nuclear dynamics. Definitely, this is a novel aspect relevant to a broad range of applications in ultra-fast physics. In particular, the careful comparison of the experimental data with the result of semi-classical calculations clearly demonstrates

the potential of holographic imaging to disentangle coupled electronic and nuclear motion.

This, in my view, justifies the publication of this manuscript in a high-visibility journal. I therefore recommend the manuscript for publication in Nature Communications after the above points have been addressed satisfactorily. A few other (minor) points listed below.

Minor points:

1. The brown rectangle in Fig. 2a is not explained in the text. I understand that it represents the spectral region used to plot the line-outs shown in Fig. 2b, but this should be mentioned explicitly.
2. After the sentence: 'The initial momenta sample the distribution function derived above for the HOMO of NO obtained from quantum-chemical calculations', it would be good to provide a reference specifying which kind of calculations were done, or which software package was used.
3. In the caption of Fig. 5, the color of the two curves in the middle panel of 5a are not referenced explicitly.
4. In the caption of Fig. 5, 'Normalized total electron the curve in the top panel of Fig. 1a is referenced as the normalized total electron yield. In the text, it is introduced as the 'angle-integrated photo-electron signal'. I guess it correspond to the yield integrated over the brown rectangle in Fig. 2a, but this is a bit confusing.

Reviewer #3 (Remarks to the Author):

The authors measured the time-resolved photoelectron-momentum images with an intense laser and report a manifestation for coupled electronic and nuclear dynamics in the NO molecules and various holographic patterns that come from electronic dynamics.

The molecular scientist must be interested in the data, in particular, the coupled dynamics, which is the first observation in the intense laser dynamics.

One of the referees recommends to publish the report in Nature Communications.

However, the following points should be considered before the publication.

- 1) The referee feels that an explanation on the coupled electronic and nuclear dynamics is too short. The authors should describe a possible mechanism of the coupled dynamics as well as the comparison with the simulated results.
- 2) Strong-field photoelectron holography and laser-induced electron diffraction are included in the observed photoelectron momentum images. The processes how holographic images and electron-diffraction patterns are extracted from the raw data should be clearly written as a whole in Methods.
- 3) The referee conjectures that the present subject is strongly related to the Λ doublet problem, which has been discussed with the conventional high-resolution molecular spectroscopy. To serve as an intermediary between the spectroscopy and the dynamics, the referee hope the authors to comment on an evolvment to studying on doublet radicals.
- 4) The bibliography should be unified.

We thank all three referees for their careful reading of the manuscript and constructive criticism, which is addressed, point by point, in the following paragraphs. Our replies are given in blue and the changes made to the manuscript in red. These changes are additionally highlighted in red in the resubmitted manuscript.

Reviewers' comments:

Reviewer #1 (Remarks to the Author):

In this manuscript, the authors study electronic and nuclear dynamics in NO molecule, by means of laser-induced electron diffraction (LIED). This powerful technique was previously used to study either static atomic/molecular systems or nuclear dynamics in molecules. Here the authors are interested in observing the coupling between both "motions" in the region of alignment revival.

The manuscript comprises two parts. They first look at the molecule when there is no "fast" nuclear motion and could determine, with the LIED, the signature of the purely electronic motion. A comparison with theory is in rather good agreement with the experimental results. In the second part, they concentrate on the "coupled region" and find strong modifications of the measured signals due to the interplay between the dynamics. Once again theoretical results support the experimental observations. While the sum of the results presented here are very interesting for the community, I cannot recommend this manuscript for publication in Nature Communications in its present form for the following reasons.

1/ I think that the techniques presented here lack the novel character. Both on the side of theory (the use of classical trajectories to simulate the diffraction signal) or experiment (impulsive alignment of molecules and holography), they were developed and presented elsewhere by other teams. The originality of this work is mainly contained in its subject and the results are convincing.

In our view, the most important innovations of our work are the following:

- 1) We report the first measurement of electronic dynamics using holography (not LIED). Previous work has only reported time-independent holography in static systems and ionization-induced nuclear dynamics. This is a significant advance, as we argue in the introduction, because it provides *structural* information on electronic wavepackets, which is not available in most other measurements. Hence, we believe that holography of electronic wave packets is really an important new technique, which has never been discussed before.
- 2) We also report the first study of coupled electronic-nuclear dynamics using any strong-field method, as remarked by referees 2 and 3. This is also an important step forward because electron-nuclear coupling is an ubiquitous phenomenon in polyatomic molecules and plays a central role in excited-state dynamics and photochemistry. This advance therefore opens up new possibilities for strong-field methods such as strong-field ionization, LIED and holography.

2/ I believe that this manuscript is not really well written. It is sometimes hard to follow the flow of thinking and some information is scattered in different parts. For example it is difficult to understand how the results of Fig. 2b are obtained. I suggest that the authors define a quantity $S(t_1, t_2) = [2(D(t_1) - D(t_2))] / (D(t_1) + D(t_2))$, and use it throughout the paper to characterize the quantities plotted.

We thank the referee for this suggestion. We have followed this recommendation.

We have added the proposed definition on p.2 and have used it in the captions of Figs. 1, 2 and 5.

3/ I have few questions concerning some results presented here.

a/ As the electronic oscillation period is 270fs, I do not understand why the difference in time between the max (1.72ps) and the min (1.56ps) is 160fs corresponding to a period of 320fs? The same applies to the other timings with a difference of 190fs.

We thank the referee for this pertinent comment. The period of the electronic dynamics that is expected on the basis of the energy-level separations is indeed 277 fs, which is identical to the value determined from the sinusoidal fit (red line in Fig. 1a). A closer look at the NO⁺ signal in Fig. 1a and 5a (central panel) however shows that the measured signal does not exactly follow the red curve, but shows additional structure, which originates from the fact that each of the two electronic states has its own rotational fine structure. We have therefore chosen the delays corresponding to the actual minima and maxima of the NO⁺ signal, which simultaneously correspond to minima and maxima of the total electron signal, to record the high-quality momentum images used in our analysis. The difference between these delays therefore differs from the nominal period of the electronic dynamics. A second factor contributing to this deviation is the finite delay-step size of 25 fs, which was chosen to keep the duration of a single measurement below 24 hours.

b/ In Fig. 1a, The authors schematically represent the oscillation of the wavefunction and I suppose that the color coding would correspond to its sign. I think it is misleading as it is the density which is oscillating (rotating) and not the wavefunction or the latter acquires a phase and becomes complex. I think the authors should be more precise on that point. The same remark applies to Fig. 3a.

We thank the reviewer for correctly pointing out that we had not carefully explained the images of the rotating wave functions in Fig. 1 and Fig. 3.

We have extended the caption of Fig. 1, part a, which now reads as follows:

“The top panel illustrates the prepared electronic and rotational dynamics with the left part showing the time evolution of the one-electron wave function in NO and the right part illustrating the molecular-axis distribution. The wave function can be chosen real-valued with opposite signs (red and blue) by multiplication with a global phase factor that corresponds to a physically irrelevant shift of the absolute energy scale. The middle panel shows the time-dependent normalized NO⁺ yield (blue line) and a sine function with a period of 277 fs (red line). The lowest panel shows the calculated alignment dynamics expressed by $\langle \cos^2 \theta \rangle$ (including volume averaging).”

In the caption of Fig. 3 we have replaced "one-electron density of NO" by "one-electron wavefunction" and furthermore we have replaced "cut through the vertical plane" by "cut through the density along the vertical plane".

c/ In Fig. 2a, the authors added the white and black lines to indicate the max and the min of the interferences. While the agreement is good for the two first lines, there is a shift for the others. Is there a simple explanation? I believe the authors should comment on that point. Furthermore to increase the visibility of these fringes for the reader, I would suggest to the authors to add these black/white lines, only on the left part of the figure.

We thank the referee for this important comment. The discrepancy between the positions of the maxima/minima in the experimental data and the simple model from Ref. [37] is caused by the

absence of the Coulomb potential in the latter. This can be shown by comparing our semi-classical trajectory calculations, which include the Coulomb interaction, with the model from Ref. [37], shown in the revised Fig. 4c. The same discrepancies can be observed in the two comparisons, i.e. the first minimum of the model (white) coincides with the first off-center maximum (in red) of the experimental data (Fig. 2a) or model (Fig. 4c) and the first off-center maximum of the model (black) coincides with the second off-center minimum of the experiment (yellow) and model (blue). Since the semi-classical trajectory calculations only include a (single-center) Coulomb potential and no short-range potential, we can directly conclude that the main source of discrepancy between the experiment and the simple model is the lack of Coulomb potential in the latter.

We have added the model predictions to Fig. 4c, the following text to p.3, left col., par. 1:

“The lack of quantitative agreement between the experiment and the simple model mainly originates from the absence of the Coulomb potential in the latter, as we further discuss below.”

And the following text on p.4, right col., par. 2:

“Interestingly, the comparison of these calculations with the simple model from Ref. \cite{bian11a} shows the same discrepancies as Fig. 2a, i.e. the model overestimates the spacing of the holographic fringes along the lateral-momentum direction. The improved agreement between the experiment and our trajectory calculations must therefore originate from the inclusion of the Coulomb potential in the latter.”

We have also followed the second recommendation of the referee and are now showing the predictions of the model only in the top right quarter of Figs. 2a and 4c.

d/ I am troubled by the 2nd paragraph of p.3. The authors claim that there is almost no angular variation of the signal in Fig2.a in this region, where I can clearly see stripes with changes of around 50% (red and black). Is this due to the detection process? Furthermore, I do not understand the conclusion of this paragraph, and think that the authors should develop a little their point.

We thank the referee for this comment. In our view, what the reviewers calls "changes of around 50% (red and black)" are really changes between red and noise. The red color corresponds to the central value of the color scale while the noise oscillates between black and white (maximum and minimum of the color scale). We would like to recall at this point that the noise increases as one moves away from the center, and due to the normalization process the noise varies between black and white. From this, we conclude that the noisy signal is indeed at a similar average value as the red parts. The vertical line that is visible in the image originates from a known artefact of the inversion procedure (so called “center noise”), which becomes visible at small signal levels. This leads to our conclusion that the angular variation is small in the high-energy part.

We have added this sentence about the noise:

“We note that the noise along the central vertical line is an artifact of the Abel inversion procedure.”

We have clarified the conclusion of the paragraph on page 3 by replacing the last sentence with the following two sentences:

“We thus conclude that the electronic dynamics in NO causes a variation of the rescattering probability but it hardly affects the shape of the photoelectron distribution in the rescattering region. In contrast, the shape of the holographic pattern shows a strong signature of the electronic dynamics. “

e/ In Fig. 4, the authors present holography signal from classical trajectory calculations. It is very hard to quantify the agreement with the experiment, in particular the periodicity of the fringes, as they are not shown in the same frame.

We have added the predictions of the simple model from Ref. [37] to Fig. 4c. These black-and-white lines can now be used as a ruler to compare the experimental data in Fig. 2a and the trajectory calculations in Fig. 4c. This comparison shows that the experimental data and the trajectory calculations are in very good agreement.

f/ Commenting Figs. 5, the authors state that the agreement between theory (e) and experiment (f) is very good. While I agree on the shape of the curves, I seriously question the way they present their results. In particular the abscissa range are not the same!! It is from -0.2 to 0.2 for the simulations and -0.05 to 0.15 for the experiments. The problem is that by making this choice, the reader miss the fact that the curves never cross zero in the experiments which they do in the simulations. I would prefer that the authors use the same frame and comment on the discrepancy eventhough there would be a qualitative agreement.

We have also followed this recommendation and are now showing theory and experiment on the same scale in Fig. 5 e and f.

We have added the following text on p.6, left col., par. 1:

“The calculation however overestimates the contrast of the normalized difference, which can originate from the limited accuracy of the electron-molecule scattering calculations and/or the strong-field ionization rates. We have verified that uncertainties in the molecular axis distributions cannot explain the observed discrepancy.”

g/ I could not find the values of t_1, t_2, t_3 and t_4 in the manuscript.

We have added these values to the caption of Fig. 5.

Reviewer #2 (Remarks to the Author):

Referee report on the manuscript NCOMMS-16-29217-T: ‘Dynamics of valence-shell electrons and nuclei probed by strong-field holography and rescattering’ by Samuel G. Walt et al.

The authors report on an experimental study where they investigate the electronic and nuclear (rotational) dynamics that are taking place in NO molecules subjected to a strong femtosecond laser pulse. They use an electron imaging spectrometer operated in the velocity mapping mode to measure the final momentum distribution of the photo-electrons as a function of the delay between a pump pulse that initiates the dynamics and a probe pulse that photo-ionizes the NO molecule.

By investigating the normalized difference of momentum distributions recorded at different pump probe delays, the authors demonstrate that, depending on the delay, the observed (holographic) interference pattern is dominated either by the electronic dynamics alone, or by the coupled electronic and rotational dynamics.

The manuscript is well written, and the results include some novel and interesting aspects. My main criticism is that earlier original work on the subject is not properly cited:

A. In their work, the authors use essentially the same experimental technique and methods as in Refs [PHYSICAL REVIEW A 87, 031404(R) (2013)] and [PHYSICAL REVIEW A 88, 023423 (2013)], where the electron motion in the ground state of the carbon atom, and the dynamics of orbital alignment in the

ground state of several atoms were time resolved. In view of these earlier studies (that have to be cited here), the authors need to revise one of their main claims, that is that holographic imaging has remained restricted to static systems and has never been applied before to the study of (bound) electronic dynamics. The new aspect of the present study thus only resides in the identification of the *coupled* electronic and nuclear motion.

B. I also would like to bring to the authors attention that (holographic) interference of different electron trajectories emerging on either side of an atom or ion during strong field photo-ionization was predicted and discussed theoretically long time ago (see for instance Ref. [gribakin97a, Phys. Rev. A 55, 3760 (1997)]). It was shown there, how the symmetry of the electron wave function determines the relative phases of the interfering trajectories. Experimentally, the first direct experimental observation of the interference effect (i.e. the holographic interference pattern), succeeded in experiments on strong field photo-detachment of negative ions in Ref. [reichle01a, Phys. Rev. Lett. 87, 243001 (2001)] (i.e. 10 years before the publication of Ref. [22]). The interference effect and its dependence on the symmetry of the initial state have been investigated in detail in a series of studies (see for instance Refs [reichle03a, J. Mod. Opt. 50, 461, (2003)] and [bergues07a, Phys. Rev. A 75, 063415 (2007)]). Even though a different nomenclature was used, (i.e. the term 'interference effect' rather than the term 'holography'), the effect is the same. These original studies have to be properly cited in the context of strong field electron holography.

We do agree with the reviewer that interference between electron paths was already built in the Keldysh-type theories, as nicely explained in the article by Gribakin and Kuchiev PRA 55, 3760 (1997), and such interference was beautifully demonstrated experimentally in the work on photodetachment from negative ions, PRL 87, 243001 (2001) and later papers. However, we emphasize that "photoelectron holography" in the sense introduced by Huisman et al., Science (2011), and used in our manuscript refers to the interference between scattered electrons and reference electrons. Since the Keldysh-type theory by Gribakin and Kuchiev does not include rescattering, it does not include holography in the present sense. The interference discussed in the earlier publications is between electron waves launched in *different* quarter cycles of the laser field. This type of interference is sometimes termed as intracycle interference. In contrast, the typical holographic signature (stripes in the momentum distribution that are roughly parallel to the polarization axis), is due to interference between electron waves in the *same* quarter cycle.

To acknowledge the earlier work on interference and on electronic dynamics in atoms, and to clarify the relation to our results, we have replaced the last sentence of the first paragraph, previously formulated as "Electronic dynamics have so far escaped observation by any of these powerful methods." by the following text:

"Although the imprint of electronic dynamics on photoelectron distributions has been reported for neutral atoms [Hultgren et al 2013], it is not obvious whether the distributions can be interpreted in terms of LIED or holography. As far as molecules are concerned, electronic dynamics have so far escaped observation by any of these powerful methods. Since LIED and holography rely both on the wave nature of the electron and on rescattering, they are expected to be sensitive to small variations of the molecular structure. Earlier work on electron interference without rescattering [Gribakin and Kuchiev PRA 1997, Reichle et al. PRL 2001, Lindner et al. PRL 2005, Bergues et al. PRA 2007] has already revealed the sensitivity to the symmetry of the bound state."

Having pointed this out, I find the results of the present study particularly interesting. By extending the methodology of Ref. [PHYSICAL REVIEW A 87, 031404(R) (2013)] to a molecular target, the authors could investigate the effect of *coupled* electronic and nuclear dynamics. Definitely, this is a

novel aspect relevant to a broad range of applications in ultra-fast physics. In particular, the careful comparison of the experimental data with the result of semi-classical calculations clearly demonstrates the potential of holographic imaging to disentangle coupled electronic and nuclear motion.

This, in my view, justifies the publication of this manuscript in a high-visibility journal. I therefore recommend the manuscript for publication in Nature Communications after the above points have been addressed satisfactorily. A few other (minor) points listed below.

Minor points:

1. The brown rectangle in Fig. 2a is not explained in the text. I understand that it represents the spectral region used to plot the line-outs shown in Fig. 2b, but this should be mentioned explicitly.

We have added this information to the caption of Fig. 2b:

“Lineouts of the holographic pattern obtained by integrating the signal inside the grey box in panel a over p_{z} , but recorded at the indicated delays that sample one period of the electronic dynamics and are referenced to a delay of 1.05 ps.”

2. After the sentence: ‘The initial momenta sample the distribution function derived above for the HOMO of NO obtained from quantum-chemical calculations’, it would be good to provide a reference specifying which kind of calculations were done, or which software package was used.

We have added this information on p.4:

“using the GAMESS package.”

3. In the caption of Fig. 5, the color of the two curves in the middle panel of 5a are not referenced explicitly.

We have added this information to the caption of Fig. 5a.

4. In the caption of Fig. 5, the curve in the top panel of Fig. 1a is referenced as the normalized total electron yield. In the text, it is introduced as the ‘angle-integrated photo-electron signal’. I guess it correspond to the yield integrated over the brown rectangle in Fig. 2a, but this is a bit confusing.

The displayed signal is averaged over all momenta, not just those within the rectangle of Fig. 2a.

We have clarified this information by replacing both occurrences with the wording

“photoelectron yield integrated over all momenta”.

Reviewer #3 (Remarks to the Author):

The authors measured the time-resolved photoelectron-momentum images with an intense laser and report a manifestation for coupled electronic and nuclear dynamics in the NO molecules and various holographic patterns that come from electronic dynamics.

The molecular scientist must be interested in the data, in particular, the coupled dynamics, which is the first observation in the intense laser dynamics.

One of the referees recommends to publish the report in Nature Communications.

However, the following points should be considered before the publication.

1) The referee feels that an explanation on the coupled electronic and nuclear dynamics is too short. The authors should describe a possible mechanism of the coupled dynamics as well as the

comparison with the simulated results.

The coupling of electronic and rotational degrees of freedom can lead to a variation of the amplitude of the electronic dynamics, causing for example an enhancement of the electronic beat around 8.5ps and a suppression around 9.5ps (see Fig. 1a). For the detailed analysis of the electron-momentum distributions, however, we focus on the region of delay times around 5ps where both electronic and rotational dynamics are causing large signal changes. The calculation of the rotational dynamics (which includes both populated spin-orbit states, as well as all couplings except for lambda doubling) predicts that the strongest antialignment (4.85ps, close to t_1) occurs slightly before the minimum of the NO⁺ yield (at 4.9 ps, close to t_2). This is in agreement with the observation that the normalized difference between t_1 and t_2 in Fig. 5b shows the same overall holographic structure as the other normalized difference plots, but the positions of extrema in the substructure on top of the holographic maxima are inverted compared to Figs. 5c,d. Unfortunately, we have to limit our discussion to these considerations because the development of a complete theoretical model for these effects would constitute a huge challenge, which lies beyond the scope of this work.

We have added the following text on p. 5, left column, par. 1:

“These calculations have been done with the methods described in Refs. \cite{baykusheva14a,zhang15a}. They include the rotational levels of both spin-orbit components of the electronic ground state of NO and therefore fully account for the coupled electronic and rotational dynamics.”

We have rewritten the discussion of the coupled electronic and nuclear dynamics on p. 5, right column, by integrating the arguments given above:

“We first note that the minimum of the photoelectron and NO⁺ yield (t_2) lies between the minima of $\langle \cos^2 \theta \rangle$ (close to t_1) and that of the extrapolated purely electronic quantum beat (close to t_3). This is a first that both electronic and nuclear dynamics influence the ionization probability. Much more details are visible in the photoelectron images. We focus our discussion on three aspects of these images.

First, Figs. 5b-d all display holographic signatures similar to those of Fig. 2a. These features coincide with the location of the measured main holographic maxima in Fig. 2a, which are reproduced here as black dashed lines. The excellent agreement shows that the structures observed in Figs. 5b-d also correspond to photoelectron holography. These holographic fringes are not unexpected because all three intervals sample substantial fractions of an electronic quantum beat.

Second, each of these 6 main holographic fringes displays an additional extremum in the radial dimension, which was not observed in Fig. 2a. Comparing Figs. 5b with 5c and 5d, we observe a maximum along the radial dimension in the case of Fig. 5b and a minimum in Figs. 5c and 5d. These radial extrema cannot be explained by purely electronic dynamics because of the differences observed in comparison with Fig. 2a. We note that normalized-difference images evaluated between delays sampling fractions of the purely electronic quantum beat in the range of 1-2 ps all look qualitatively identical to Fig. 2a (not shown). The additional structures on top of the holographic fringes can therefore only originate from the simultaneous effect of electronic and nuclear dynamics, which is thus found to leave a characteristic imprint on time-dependent photoelectron holography.”

2) Strong-field photoelectron holography and laser-induced electron diffraction are included in the observed photoelectron momentum images. The processes how holographic images and electron-diffraction patterns are extracted from the raw data should be clearly written as a whole in Methods. We have added the following text to the Methods section on p. 7:

“The two-dimensional photoelectron momentum images were processed as follows. A constant background was subtracted from the entire image to account for dark counts and residual stray light. Then, the images were symmetrized. Finally, two-dimensional momentum distributions were obtained by iterative Abel inversion using a computer program developed in our group [cite{walt15b}]. A typical photoelectron momentum distribution is shown in Fig. 1b. The momentum distributions shown in Figs. 1b, 1c, 2a, and 5 b-d are normalized differences between distributions measured at two different pump-probe delays, as defined in the text. The analysis of the LIED signals proceeded as follows. For a given photoelectron momentum at rescattering \vec{p}_s , the corresponding time of rescattering t_s was determined using the long trajectories in the classical recollision model (see e.g. Ref. [cite{lin10a}]). The short trajectories were discarded because they are suppressed by the lower strong-field ionization rate at the corresponding times of ionization. The additional momentum shift \vec{p}_L that the electrons acquire from the laser field after rescattering was calculated according to $\vec{p}_L = -\vec{A}(t_r)$, using the vector potential corresponding to the laser field used in our experiment.”

3) The referee conjectures that the present subject is strongly related to the Λ doublet problem, which has been discussed with the conventional high-resolution molecular spectroscopy. To serve as an intermediary between the spectroscopy and the dynamics, the referee hope the authors to comment on an evolvement to studying on doublet radicals.

We thank the referee for this thoughtful comment. Lambda doubling is indeed a very interesting phenomenon that occurs in all electronically degenerate states of linear molecules. It is a consequence of the coupling between electronic and rotational angular momenta. In NO, lambda doubling splits each rotational level of both spin-orbit states into two sub-levels of opposite parities (see Fig. R1, taken from the supplementary material of Ref. [8]). These splittings are extremely small, ranging from 10^{-4} to 10^{-1} cm^{-1} (R. N. Zare, Angular Momentum (John Wiley & Sons, New York, 1988)). The associated dynamics therefore takes place on time scales from 330 ns to 330 ps, which are well beyond the time scales studied in our work. We will carefully consider this interesting idea for a future experiment.

Fig. R 1: Level diagram of the two lowest electronic states of nitric oxide. Each rotational state is split into two levels of opposite total parity by Λ -doubling. The separation of the two Λ -doublets (in the range of 10^{-4} to 10^{-1} cm^{-1}) is largely exaggerated in the figure and its effect is negligible in the present study.

4) The bibliography should be unified.

We have done this.

REVIEWERS' COMMENTS:

Reviewer #1 (Remarks to the Author):

The authors complied with the recommendations of the other referees and myself. Thus, I think the manuscript gained a lot in readability and is more sound. Therefore I recommend its publication in Nature Communications.

Reviewer #2 (Remarks to the Author):

In my opinion, the points raised by the referees have been satisfactorily addressed in the revised version of the manuscript. My recommendation is therefore the publication of the manuscript in its present form.

Reviewer #3 (Remarks to the Author):

The authors appropriately revised the manuscript with respect to Comment 1 and 2. The revised version has become explicit over the whole manuscript. As concerns the comment 3, the referee's message was less apparent so it seems that the authors' attention is paid to the rotational levels. The referee simply meant that it might be worth comparing between phenomena that occurs the different relative rotational axes for the molecule and the orbital. But the referee understands that such discussion is hard at the present stage as written in the reply for Comment 1. Conclusively, the reviewer #3 recommends to publish this report in Nature Communications.